

# Difference equations: From Berry connections to the Coulomb branch

### Andrea E. V. Ferrari[1,2*] and Daniel Zhang[3,4†]

**1** School of Mathematics, The University of Edinburgh, Mayfield Road, Edinburgh, U.K.
**2** Deutsches Elektronen-Synchrotron DESY, Notkestr. 85, 22607 Hamburg, Germany
**3** Mathematical Institute, University of Oxford, Woodstock Road, Oxford, U.K.
**4** St John's College, University of Oxford, St Giles', Oxford, U.K.

⋆ andrea.e.v.ferrari@gmail.com , † daniel.zhang@sjc.ox.ac.uk

## Abstract

In recent work, we demonstrated that a spectral variety for the Berry connection of a 2d $\mathcal{N} = (2,2)$ GLSM with Kähler vacuum moduli space $X$ and Abelian flavour symmetry is the support of a sheaf induced by a certain action on the equivariant quantum cohomology of $X$. This action could be quantised to first-order matrix difference equations obeyed by brane amplitudes, and by taking the conformal limit, vortex partition functions. In this article, we elucidate how some of these results may be recovered from a 3d perspective, by placing the 2d theory at a boundary and gauging the flavour symmetry via a bulk A-twisted 3d $\mathcal{N} = 4$ gauge theory (a sandwich construction). We interpret the above action as that of the bulk Coulomb branch algebra on boundary twisted chiral operators. This relates our work to recent constructions of actions of Coulomb branch algebras on quantum equivariant cohomology, providing a novel correspondence between these actions and spectral data of generalised periodic monopoles. The effective IR description of the 2d theory in terms of a twisted superpotential allows for explicit computations of these actions, which we demonstrate for Abelian GLSMs.

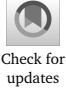

# 1  Introduction

The study of relations between observables in supersymmetric quantum field theories (QFTs) enjoying an effective description governed by some geometric space $X$, and cohomology theories of $X$, has a long and rich history. Recently, in the context of QFTs endowed with a flavour symmetry $T = U(1)^n$, the geometry of Berry connections over deformation parameters for $T$ has been demonstrated to be a useful tool to push this relation further in several distinct directions [1–4]. In this article, we elaborate upon elements of [3,4] and uncover relations with other remarkable topics.

The starting observation of [3] was that Berry connections for 2d GLSMs $\mathcal{T}_{2d}$ quantised on a circle and with such a flavour symmetry $T$ are generalised periodic monopoles of Cherkis-Kapustin type [5,6]. It was then observed that if $\mathcal{T}_{2d}$ flows to a non-linear sigma model with Kähler target $X$, the spectral variety of this monopole is the support of a sheaf defined by an action of a certain algebra on $QH_T^\bullet(X)$. The algebra could be identified with the algebra of functions on a 3d $\mathcal{N} = 4$ Abelian Coulomb branch $\mathbb{C}[\mathcal{M}_C]$. Moreover, building on work relating periodic monopoles to certain difference modules [7], the variety could be quantised by certain first-order matrix difference equations for brane amplitudes or, in the conformal limit, vortex or hemisphere partition functions.

The main goal of this short article is to show that the identification of the above algebra with $\mathbb{C}[\mathcal{M}_C]$ can be taken seriously, and fruitfully so. The basic idea, inspired by previous works in physics and mathematics [8,9], is to represent the twisted 2d GLSM (which for simplicity we take to be Abelian and with compact target) in terms of a so-called sandwich construction.[1]

---

[1]In the generalised symmetry literature, these sandwich constructions have recently gained popularity, see [10] for a seminal paper.

More specifically, we consider (A-twisted) 2d GLSMs and represent them in terms of an (A-twisted) 3d $\mathcal{N} = 4$ pure gauge theory with gauge group $T$ on a geometry $I \times \mathbb{R}^2$ where $I$ is an interval at whose ends two distinct 2d $\mathcal{N} = (2, 2)$ boundary conditions are prescribed:

- "Right boundary": the 3d vector multiplet is given Neumann boundary conditions, enriched by the presence of a 2d theory $\mathcal{T}_{2d}/T$ obtained from $\mathcal{T}_{2d}$ by gauging $T$.

- "Left boundary": the 3d vector multiplet is given a Dirichlet boundary condition, fixing the complex scalar $\varphi$ (in a particular choice of complex structure) to a fixed value, and the gauge symmetry $T$ is broken back to a global symmetry $T$.

Since the length of the interval $I$ is exact, one can squash it to recover the original GLSM. The setup can further be subjected to an Omega background.

The 3d description is illuminating in that the bulk Coulomb branch operators can be made to act on either boundary, with actions that encode the ones encountered in the description of the Berry connection spectral data. In fact, by requiring a consistent action on the left and right boundary, we demonstrate that

- In the absence of the Omega deformation, the bulk Coulomb branch operators can be brought to the boundary to act on the twisted 2d chiral ring, which can be identified with $QH_T^\bullet(X)$. The boundary twisted chiral ring therefore forms a module over the bulk Coulomb branch algebra, with support on the image of the boundary condition, which coincides with the spectral variety of the Berry connection.

- In the presence of the Omega deformation, the above statement is quantised, and the space of boundary twisted chiral operators determines a module for the *quantised* Coulomb branch algebra $\hat{\mathbb{C}}_\epsilon[\mathcal{M}_C]$. Consistency of the action on the left and right boundaries leads to difference equations equivalent to those derived in [3, 4].

Our motivation to study this alternative interpretation of spectral varieties and difference equations is at least twofold. On the one hand, we demonstrate very concretely why actions of Coulomb branch operators on the equivariant quantum cohomology of the target of some 2d (2, 2) GLSM are related to spectral data of generalised periodic monopoles. Thus, the physical setup provides an explicit relationship between two so far distinct important topics in mathematics.[2] On the other hand, the novel perspective opens the way to different computational techniques, mainly due to the fact that in the infrared the theory is controlled by an effective twisted superpotential $\widetilde{W}_{\text{eff}}$. This can in turn be used to build spectral varieties for generalised periodic monopoles that arise as Berry connections, as well as their quantisations.

This new vantage point is amenable to obvious generalisations. Having discussed how to couple the 2d GLSMs to a bulk pure gauge theory, it is only natural to ask what happens when the theory is coupled to more general 3d gauge theories. Such questions are related to actions of Coulomb branch operators on $QH_T^\bullet(X \times L)$ where $X$ is the compact target of the GLSM and $L$ is some affine space, which are currently being discussed in the mathematical literature [9, 12–14]. In this article, we demonstrate how one can consistently consider such richer setups and we work out some examples based on physical intuition and techniques. However, we conclude that for the sake of the difference equations considered here, these more general setups do not add further essential ingredients.

---

[2]Notice that spectral data for generalised periodic monopoles is in particular part of the holomorphic Floer theory programme of Kontsevich and Soibelman [11]. See [3] for some remarks on other connections of this to generalised cohomology theories.

**Further directions.** An obvious direction for further investigation is to consider what happens when one replaces either the bulk or boundary theory by one with a *non*-Abelian and gauge group. Generalising $\mathcal{T}_{2d}$ to a non-Abelian GLSM should be relatively straightforward, and the nature of the generalisation is mostly technical.[3] However, the interpretation of this set-up generalised to a non-Abelian *bulk* gauge group from the point of view of a Berry connection is not clear, and we hope to return to this interesting question in future work.[4] Finally, as noted already in [3], we expect that the difference equations studied in this work to arise as a 2d limit of the qKZ equations obeyed by vertex functions *i.e.* holomorphic blocks *i.e.* hemisphere partition functions in 3d. Those equations should also admit an interpretation as an action of bulk 4d operators on those of a 3d theory living on its boundary.

**Summary.** This article is structured as follows. In Section 2 we review the setup of this work, as well as the relation between spectral data of Berry connections and difference equations for vortex or hemisphere partition functions. In Section 3 we explain how the spectral variety of the Berry connections can be identified with the image of the right boundary condition described above inside the 3d Coulomb branch. In Section 4 we introduce an Omega background and explain in detail how the (enriched) boundary conditions now define modules for the quantised bulk Coulomb branch algebra, and how the difference equations can be extracted from these modules. In Section 5 we explain how these results can be extended to more general 3d $\mathcal{N}=4$ gauge theories with matter coupled to boundary 2d $\mathcal{N}=(2,2)$ GLSM. Throughout the article, we discuss in detail the example of SQED[$N$], or the $\mathbb{CP}^{N-1}$ sigma model, whose associated Berry connection is a smooth $SU(N)$ periodic monopole [3].

## 2 Berry connections & difference equations

Let $\mathcal{T}_{2d}$ be a 2d GLSM with gauge group $G$ and an Abelian flavour symmetry $T \cong U(1)^n$. For convenience, we often take the GLSM to be Abelian, but much of the ensuing discussion applies to non-Abelian GLSMs as well. We suppose there is an IR phase in which the GLSM flows to a nonlinear sigma model (NLSM) to some Kähler variety $X$, which unless otherwise stated we will take to be compact. We can introduce a twisted complex mass $m \in \mathfrak{t}_{\mathbb{C}} \cong \mathbb{R}^{2n}$ for $T$, with components that we will denote by $m_i$, and assume that at a generic value of $m$ the theory has isolated, massive vacua. In a different phase, the vacua are determined by the low energy effective twisted superpotential $\widetilde{W}_{\text{eff}}(\sigma, m)$, which is a function of the (Abelianised) complex scalar $\sigma$ in the Cartan of $G$, and $m$. This object turns out to be key in our constructions, since the twisted chiral ring is renormalisation invariant and thus may be determined explicitly in terms of $\widetilde{W}_{\text{eff}}(\sigma, m)$.

### 2.1 Spectral variety & quantum cohomology

Consider now the 2d GLSM $\mathcal{T}_{2d}$ on a cylinder $S^1 \times \mathbb{R}$. Besides the complex mass $m \in \mathbb{R}^{2n}$, one can introduce an $(S^1)^n$-valued holonomy, resulting in the space of deformation parameters

$$M \cong (S^1 \times \mathbb{R}^2)^n. \tag{1}$$

We will henceforth introduce coordinates $(t_i, w_i)$ on each copy of $S^1 \times \mathbb{R}^2$, where $w_i = -2im_i$.

---

[3]In particular, the boundary twisted chiral ring (and quantum K-theory of $X$) is modified to symmetric polynomials in the (boundary) gauge fugacities modulo the Bethe/vacuum equations.

[4]During the completion of this work, we have been made aware of upcoming work on these sandwich constructions with a non-Abelian bulk gauge group based on extensions of quasi-maps moduli spaces [15]. It would be very interesting to investigate whether these new approaches can shed light on such questions.

The space of supersymmetric ground states in $Q_A$ cohomology forms a bundle

$$E \to M \tag{2}$$

of rank $N$. This is endowed with a Hermitian metric $h$, unitary connection $A_i$ and anti-Hermitian operator $\phi_i$. The $tt^*$ equations obeyed by the Berry connection are [16]

$$
\begin{aligned}
\left[\bar{D}_i, C_j\right] &= 0 = \left[D_i, \bar{C}_j\right], \\
\left[D_i, D_j\right] &= 0 = \left[\bar{D}_i, \bar{D}_j\right], \\
\left[D_i, C_j\right] &= \left[D_j, C_i\right], \quad \left[\bar{D}_i, \bar{C}_j\right] = \left[\bar{D}_j, \bar{C}_i\right], \\
\left[D_i, \bar{D}_j\right] &= -\left[C_i, \bar{C}_j\right],
\end{aligned}
\tag{3}
$$

where

$$
\begin{aligned}
D_i &= \partial_{w_i} - A_{w_i}, & \bar{D}_i &= \bar{\partial}_{\bar{w}_i} - \bar{A}_{\bar{w}_i}, \\
C_i &= D_{t_i} - i\phi_i, & \bar{C}_i &= -D_{t_i} - i\phi_i,
\end{aligned}
\tag{4}
$$

and $\phi_i$ is an anti-Hermitian adjoint Higgs field. In the $n = 1$ case, these equations become

$$F = \star D\phi, \tag{5}$$

and so the Berry connection is a generalised periodic monopole on $M$.

By working in the cohomology of the supercharge $Q_A$ we have picked a so-called mini-complex structure on each copy $(S^1 \times \mathbb{R}^2)$ of $M$. This implies in particular that $(S^1 \times \mathbb{R}^2)$ is endowed with a complex structure on $\mathbb{R}^2 \cong \mathbb{C}$. For later convenience, we denote the space $M$ endowed with this structure by $M^0$.

### 2.1.1 Spectral variety

The bundle $E|_{t_i = t_i^*}$ restricted at a fixed value of all of the $t_i$, say $t_i = t_i^*$, has $n$ commuting Dolbeault operators $\partial_{\bar{w}_i}$ that endow it with a holomorphic structure. Due to (7), one can consider parallel transport of its holomorphic sections with respect to $\partial_{t_i}$ to obtain holomorphic sections at different values of the $t_i$.

The equations (3) imply in particular that the operators

$$\partial_{t_i} := D_{t_i} - i\phi_i, \qquad \partial_{w_i} := D_{w_i}, \tag{6}$$

satisfy

$$\left[\partial_{t_i}, \partial_{\bar{w}_j}\right] = 0. \tag{7}$$

This endows the bundle with a so-called mini-holomorphic structure [7]. Thus, we can parallel transport holomorphic sections at $t_i = 0$ around any of the $n$ cycles $S^1$. This defines $n$ commuting $\mathbb{C}[w]$-linear automorphisms of $E|_{t_i=0}$, which we denote by $F_i(w_i)$.[5] It is therefore possible to define what was called in [3] the Cherkis-Kapustin *spectral variety*. This is a Lagrangian subvariety of $(\mathbb{C}^* \times \mathbb{C})^n$, which is characterised by the vanishing of $n$ rational functions

$$\mathcal{L}_i(p_i, w_i) = \det(p_i \mathbf{1} - F_i(w_i)). \tag{8}$$

One of the results of [3] is that the spectral variety $\mathcal{L}$ can be computed by starting from the vacuum (Bethe) equations of the theory (now written for convenience in terms of $m_i$)

$$e^{\frac{\partial \widetilde{W}_{\text{eff}}(\sigma, m)}{\partial \sigma_a}} = 1, \qquad a = 1 \ldots r, \tag{9}$$

---

[5]As we will comment further in Section 3, in the presence of a non-compact target one ought to consider sections that are meromorphic, and the automorphisms $F_i(w_i)$ may be rational.

and supplementing them with the equations

$$e^{\frac{\partial \widetilde{W}_{\text{eff}}(\sigma,m)}{\partial m_i}} = p_i, \qquad i = 1 \dots n,$$
(10)

where $p_i$ are the 'momenta' conjugate to $m_i$. More precisely, it was argued that upon elimination of $\sigma_a$ from this system of equations one can recover (8). Although reviewing the full argument goes beyond the scope of this article, this is essentially because the holonomies around the full circle can be computed in the large radius limit of the cylinder. In this limit, the ground states are in correspondence with the vacua of the theory. The eigenvalues of $F_i(w)$ can then be computed by evaluating an appropriate operator at the vacua. By inspecting the Lagrangian, it is possible to check that the relevant operator is given by (10).

### 2.1.2 Quantum cohomology

Under our assumptions the equations (9) correspond to the ring relations of the equivariant quantum cohomology ring $QH_T^\bullet(X)$, *i.e.* the twisted chiral ring of $\mathcal{T}_{2d}$ (the ring of operators in $Q_A$-cohomology) [17]. As argued in [3] and reviewed in [4], an interpretation of the spectral variety can be given as follows. First, notice that there is an action of the ring of functions $(w_i, p_i)$, $i \in \{1, \dots, n\}$ on $(T^*\mathbb{C}^*)^n$ on $QH_T^\bullet(X)$, where $p_i$ acts by means of the identification

$$p_i = F_i(w).$$
(11)

This action determines a sheaf over $(T^*\mathbb{C}^*)^n$, and by definition, the spectral variety corresponds to the support of this sheaf. It will be the goal of Section 3 of this article to offer another interpretation of $(T^*\mathbb{C}^*)^n$ in terms of the Coulomb branch of a pure Abelian 3d $\mathcal{N}=4$ theory, and to identify the above action with the one studied in the work [9]. Before doing so, we will briefly review a set of difference equations for hemisphere or vortex partition functions that can be obtained by deforming the supercharge $Q_A$ to a $\mathbb{P}^1$ family of supercharges $Q_\lambda$.

### 2.1.3 Example: SQED[2]

We reproduce for the reader's convenience an example from [3,4], namely SQED[2]. This is a $U(1)$ GLSM with two chiral multiplets $\Phi_1$, $\Phi_2$ of charges $(+1,+1)$ and $(+1,-1)$ under $G \times T$. In the IR this becomes a $\mathbb{CP}^1$ $\sigma$-model. The effective twisted superpotential is given by

$$\widetilde{W}_{\text{eff}} = -2\pi i \tau(\epsilon)\sigma + (\sigma+m)\left(\log\left(\frac{\sigma+m}{\epsilon}\right) - 1\right) + (\sigma-m)\left(\log\left(\frac{\sigma-m}{\epsilon}\right) - 1\right).$$
(12)

Here $\tau(\epsilon)$ is the renormalised complex FI parameter

$$\tau(\epsilon) = \tau_0 + \frac{2}{2\pi i}\log(\Lambda_0/\epsilon),$$
(13)

where $\Lambda_0$ is a fixed UV energy scale, $\epsilon$ the RG scale and $\tau_0$ the bare complexified FI parameter.

The vacuum equations describe the quantum equivariant cohomology $QH_T^\bullet(\mathbb{CP}^1)$:

$$1 = e^{\frac{\partial \widetilde{W}_{\text{eff}}}{\partial \sigma}} = q^{-1}(\sigma+m)(\sigma-m),$$
(14)

where we have defined the RG invariant combination $q = \epsilon^2 e^{2\pi i \tau_0}$. Computing the conjugate momentum

$$p = e^{\frac{\partial \widetilde{W}_{\text{eff}}}{\partial m}} \quad \Rightarrow \quad \sigma = \frac{m(p+1)}{p-1},$$
(15)

and substituting into (14) we obtain

$$\mathcal{L}(m,p) = p^2 - 2(1 + 2m^2 q^{-1})p + 1 = 0.$$
(16)

## 2.2 Difference equations for branes

The results in the previous section were based on an interpretation of the space of supersymmetric ground states in $Q_A$ cohomology as a mini-holomorphic vector bundle $E \to M^0$. In [3] a $\mathbb{P}^1$-family

$$Q_\lambda := \frac{1}{\sqrt{1 + \lambda^2}}(Q_A + \lambda \bar{Q}_A), \qquad \lambda \in \mathbb{P}^1, \tag{17}$$

of supercharges was more generally considered. One can describe the bundle of supersymmetric ground states as the vector space of states in $Q_\lambda$ cohomology, which corresponds to endowing $E \to M^\lambda$ with another mini-holomorphic structure.

More concretely, consider the parametrisation

$$(t_{0,i}, \beta_{0,i}) = \frac{1}{1 + |\lambda|^2} \left( (1 - |\lambda|^2) t_i + 2\mathrm{Im}(\lambda \bar{w}_i), w_i + \lambda^2 \bar{w}_i + 2i\lambda t_i \right). \tag{18}$$

In analogy with (6) one can define operators

$$\partial_{t_{0,i}} := D_{t_{0,i}} - i\phi_i, \qquad \partial_{\bar{\beta}_{0,i}} := D_{\bar{\beta}_{0,i}}, \tag{19}$$

which commute because of the Bogomolny equations

$$\left[ \partial_{t_{0,i}}, \partial_{\bar{\beta}_{0,i}} \right] = 0. \tag{20}$$

In a way similar to the above, one can derive $2i\lambda$-difference $\mathbb{C}(\beta_{0,i})$-modules $(\mathcal{F}_i, V)$ where

- $V$ corresponds to the sections of the bundle restricted to $t_0 = 0$ that are holomorphic with respect to $\partial_{\bar{\beta}_0, i}$, whereas

- $\mathcal{F}_i$ is defined by parallel transport with respect to $\partial_{t_0, i}$, composed with a pullback by a shift $\Phi_i : \beta_{0,i} \mapsto \beta_{0,i} + 2i\lambda$ in order to obtain a genuine automorphism of the bundle (notice there is no shift for $\lambda = 0$, as expected for consistency with the previous case).

Details of the construction of these modules and relations to the work of Mochizuki [7] on the classification of periodic monopoles can once again be found in [3,4]. Here we simply remark that the module is a $2i\lambda$-difference module since now for $|s\rangle \in V$, $f \in \mathbb{C}(\beta_{0,i})$

$$\mathcal{F}_i(\beta_0)(f|s\rangle) = \Phi_i^*(f)\mathcal{F}_i(\beta_0)|s\rangle. \tag{21}$$

Moreover, by considering states on the cylinder generated by a $D$-branes preserving $Q_\lambda$, $Q_\lambda^\dagger$ we claimed in [3,4] that one can construct states such that

$$\mathcal{F}_i |D\rangle = |D\rangle. \tag{22}$$

Picking a basis $|a^\lambda\rangle$ for the holomorphic bundle $E|_{t_{0,i}=0}$, from (22) one can derive

$$\Phi_i^* \langle a^\lambda | D \rangle = G_{ab}^{(i)}(\beta_0) \langle b^\lambda | D \rangle, \tag{23}$$

where for convenience we suppressed the dependence of $G_{ab}^{(i)}$ on $\lambda$. Here $G_{ab}^{(i)}(\beta_0)$ is a holomorphic function in $\beta_{0,i}$.

## 2.3 Difference equations for hemisphere partition functions

Instead of considering in any detail the difference equations (23), in this article, we restrict ourselves to difference equations that can be derived from those in the conformal limit. This corresponds to taking

$$\lim_c : \quad \lambda \to 0, \qquad L \to 0, \qquad \frac{\lambda}{L} = \epsilon \text{ fixed,} \tag{24}$$

where these difference equations degenerate into difference equations for hemisphere or vortex partition functions. These enjoy the additional feature that they can be computed exactly via localisation.

We denote by $\mathcal{Z}_D[\mathcal{O}_a, m]$ the hemisphere partition function on $HS^2$ [18–20], with an insertion of a twisted chiral ring operator $\mathcal{O}_a$ at the tip. The radius of $HS^2$ is identified with $\frac{1}{\epsilon}$. The boundary condition $D$ becomes a $B$-brane. In the case where all the 2d chiral multiplets are given Neumann boundary conditions, and the JK residue selects a pole associated to some vacuum $\alpha$ of the 2d theory, this coincides with the vortex partition function $\mathcal{Z}_\alpha[\mathcal{O}_a, m]$ of $\mathcal{T}_{2d}$ computed in an Omega background on $\mathbb{R}^2_\epsilon$, with a massive vacuum $\alpha$ at infinity. In either case, the difference equations (23) reduce in this limit to [3]:

$$\hat{p}_i \, \mathcal{Z}_D[\mathcal{O}_a, m] = \mathcal{Z}_D[\mathcal{O}_a, m + \epsilon e_i] = \widetilde{G}^{(i)}_{ab}(m, \epsilon) \mathcal{Z}_D[\mathcal{O}_b, m], \tag{25}$$

where $\widetilde{G}^{(i)}$, $i \in \{1, \ldots, n\}$ is a matrix of rational functions in $m$,[6] the conformal limit of the matrices $G$, and $e_i$ is a fundamental weight or unit vector in $\mathfrak{t}_{\mathbb{C}}$. To the best of our knowledge, this is a new result on difference relations satisfied by these partition functions. It was further argued that

$$\lim_{\epsilon \to 0} \widetilde{G}^{(i)}(m, \epsilon) = F^{(i)}(m). \tag{26}$$

In particular, the difference equations may be regarded as a quantisation of the spectral variety $\mathcal{L}_i(m, p) = 0$. Rather nicely, due to the exactly calculable nature of hemisphere and vortex partition functions, this gives a recipe, arising from 2d GLSMs, to construct solutions to difference equations arising as quantised spectral varieties of generalised periodic monopoles.

Note that the equations (25) are independent of the brane $D$ or vacuum $\alpha$. This suggests that they are solely a property of the chiral ring, and that a construction similar to the one in Section 5 of [21], which utilises a 3d-2d bulk-boundary system (sandwich construction) to represent a 2d GLSM, may recover them. We will show this in the subsequent sections of this article. In turn, this will allow us to make further contact with the pioneering work on actions of Coulomb branch algebras on the quantum cohomology [9].

### 2.3.1 Example: SQED[2]

Returning to the example in Section 2.1.3, we consider the following thimble branes (this is not actually important, but illustrates that the same difference equations hold for different branes):

$$D_1 : \quad \Phi_1, \Phi_2 \text{ Neumann}, \qquad D_2 : \quad \Phi_1 \text{ Dirichlet}, \qquad \Phi_2 \text{ Neumann}. \tag{27}$$

The twisted chiral ring of supersymmetric QED is generated by $\mathcal{O}_a \in \{\mathbf{1}, \sigma\}$ and is subject to the relation (14).

The hemisphere partition functions are

$$\mathcal{Z}_{D_1}[\mathcal{O}_a] = \oint_{\mathcal{C}_1} \frac{d\sigma}{2\pi i \epsilon} e^{-\frac{2\pi i \sigma \tau}{\epsilon}} \, \Gamma\left[\frac{\sigma + m}{\epsilon}\right] \Gamma\left[\frac{\sigma - m}{\epsilon}\right] \mathcal{O}_a,$$

$$\mathcal{Z}_{D_2}[\mathcal{O}_a] = \oint_{\mathcal{C}_2} \frac{d\sigma}{2\pi i \epsilon} e^{-\frac{2\pi i \sigma \tau}{\epsilon}} \, \frac{(-2\pi i) e^{\frac{\pi i (\sigma + m)}{\epsilon}}}{\Gamma\left[1 - \frac{\sigma + m}{\epsilon}\right]} \Gamma\left[\frac{\sigma - m}{\epsilon}\right] \mathcal{O}_a, \tag{28}$$

---

[6]It also depends on the Kähler parameter $\tau$, which we treat as a constant.

where $\mathcal{C}_1$ encloses the poles of $\Gamma[\frac{\sigma+m}{\epsilon}]$ at $\sigma = -\epsilon k - m$ and $\mathcal{C}_2$ encloses the poles at $\sigma = -\epsilon k + m$, where $k \in \mathbb{N}_0$. Here, $\tau = \tau(\epsilon)$ is the renormalised FI parameter (13). Note $\mathcal{Z}_{D_1}$ coincides with the vortex partition function computed on the Omega background on $\mathbb{R}^2_\epsilon$ [22]. Performing the contour integration we obtain

$$\mathcal{Z}_{D_1}[\mathbf{1}] = e^{\frac{2\pi i m \tau}{\epsilon}} \Gamma\left[-\frac{2m}{\epsilon}\right] {}_0F_1\left[1 + \frac{2m}{\epsilon}; e^{2\pi i \tau}\right],$$

$$\mathcal{Z}_{D_1}[\sigma] = -m\,\mathcal{Z}_{D_1}[\mathbf{1}] + \epsilon\, e^{2\pi i \tau} e^{\frac{2\pi i m \tau}{\epsilon}} \Gamma\left[-1 - \frac{2m}{\epsilon}\right] {}_0F_1\left[2 + \frac{2m}{\epsilon}; e^{2\pi i \tau}\right], \tag{29}$$

and

$$\mathcal{Z}_{D_2}[\mathcal{O}_a] = \left(1 - e^{\frac{4\pi i m}{\epsilon}}\right)\left(\mathcal{Z}_{D_1}[\mathcal{O}_a]\right)\big|_{m \to -m}. \tag{30}$$

Using the standard hypergeometric identity

$$ {}_0F_1(b, z) = {}_0F_1(b+1, z) + \frac{z}{b(b+1)} {}_0F_1(b+2, z), \tag{31}$$

it is not hard to check that

$$\hat{p}\begin{pmatrix} \mathcal{Z}_{D_\alpha}[\mathbf{1}] \\ \mathcal{Z}_{D_\alpha}[\sigma] \end{pmatrix} = \widetilde{G}(m, \epsilon) \begin{pmatrix} \mathcal{Z}_{D_\alpha}[\mathbf{1}] \\ \mathcal{Z}_{D_\alpha}[\sigma] \end{pmatrix}, \tag{32}$$

for both $\alpha = 1, 2$, where:

$$\widetilde{G}(m, \epsilon) = \begin{pmatrix} 1 + m(2m+\epsilon)q^{-1} & (2m+\epsilon)q^{-1} \\ (2m+\epsilon)(1 + m(m+\epsilon)q^{-1}) & 1 + (m+\epsilon)(2m+\epsilon)q^{-1} \end{pmatrix}. \tag{33}$$

It is easy to see that these equations are compatible with the classical limit (8), see [3, 4].

# 3 Coulomb branch actions & spectral varieties

In this section, we explain how to derive the results summarised above by representing the 2d GLSM $\mathcal{T}_{2d}$ in terms of a 3d sandwich construction. The basic idea [9, 21] is to gauge the flavour symmetry $T$ of the 2d GLSM and to couple it to a topologically twisted pure 3d gauge theory supporting another boundary condition where the gauge symmetry is broken back. We picture the sandwich construction in Figure 1.

Our starting point is a topologically A-twisted pure Abelian 3d $\mathcal{N} = 4$ gauge theory $\mathcal{T}_{3d}$ with gauge group $T \cong U(1)^n$. For our purposes, the topological A-twist corresponds to considering operators in the cohomology of the mirror Rozansky-Witten supercharge [23], which

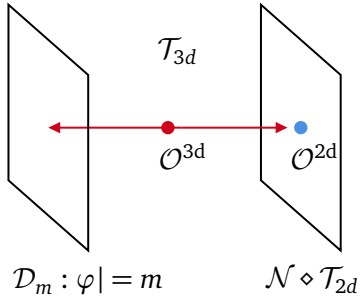

Figure 1: The sandwich construction realising the A-twisted $\mathcal{T}_{2d}$ theory as a pure 3d $\mathcal{N} = 4$ Abelian gauge theory with enriched boundary condition.

we denote $Q_C$. The Coulomb branch of pure Abelian gauge theory, whose ring of functions is in the cohomology of $Q_C$, is not quantum-corrected (see *e.g.* [24]). The ring of functions comprises half-BPS monopole operators $v_{\pm,i}$ for $i = 1,\ldots,n$, where $v_{\pm,i}$ are defined to impose that the first Chern class of the gauge bundle on a 2-sphere surrounding them is $\pm 1$. Their semi-classical expression is

$$v_{\pm,i} = e^{1/g^2(\sigma_i + i\gamma_i)}, \tag{34}$$

where $\gamma_i$ are the dual photons and $g^2$ is the coupling, and so the monopole operators satisfy

$$v_{+,i}v_{-,i} = 1. \tag{35}$$

In addition, there are $n$ complex valued scalars in the vector multiplets, which we denote as is customary by $\varphi_i$ or collectively $\varphi \in \mathbb{C}^n$. The Coulomb branch can therefore be identified with

$$\mathcal{M}_C = T^*(\mathbb{C}^*)^n = (\mathbb{C} \times \mathbb{C}^*)^n, \tag{36}$$

precisely the space in which the spectral varieties (8) are holomorphic Lagrangian subvarieties.[7]

In order to engineer the GLSM via a sandwich construction, we can consider this pure gauge theory on a geometry $I \times \mathbb{R}^2$, where $I$ is a finite interval with coordinate $t \in [0, L]$, and impose certain boundary conditions. Since the length of the interval is immaterial in the topological twist, we will be able to shrink its size to zero, leading to a 2d theory. The $(2,2)$ boundary conditions relevant to reproduce the 2d GLSM are:

- At the left boundary $t = 0$, we prescribe Dirichlet boundary conditions $\mathcal{D}_m$ for $\mathcal{T}_{3d}$, which in particular fixes the complex scalar in the vector multiplet $\varphi$

$$\varphi|_{t=0} = m. \tag{37}$$

  The remainder of the fields are fixed by supersymmetry, see [8] for further details. In particular, the components of the gauge field $A_\|$ parallel to the boundary are set to zero, and the real scalar in the 3d vector multiplet is prescribed a Neumann-like boundary condition we do not reproduce here. The (bulk) gauge symmetry $T$ is explicitly broken at this boundary, to a flavour symmetry.

- At the right boundary $t = L$, we give Neumann boundary conditions $\mathcal{N}$ for the 3d vector multiplet of $\mathcal{T}_{3d}$. In the absence of any boundary matter, this leaves $\varphi$ unconstrained at the boundary, but fixes the monopole operators[8]

$$v_\pm|_{t=L} = 1. \tag{38}$$

  We will instead place a 2d theory $\mathcal{T}_{2d}/T$ at this boundary that is obtained from the GLSM $\mathcal{T}_{2d}$ by gauging the flavour symmetry $T$ via the 3d vector multiplet. We will refer to this enriched Neumann boundary condition as $\mathcal{N} \diamond \mathcal{T}_{2d}$.

  Flowing to the IR, we thus have a $(2,2)$ Neumann boundary condition for the gauge multiplet coupled (in the IR description) to a boundary 2d twisted chiral multiplet valued in the adjoint of the 2d gauge group, with bottom component $\sigma$, via an effective twisted 2d superpotential $\widetilde{W}_{\text{eff}}(\sigma, \varphi)$. This is the same superpotential as before but with $m$ replaced with $\varphi$.

---

[7]The symplectic structure on the Coulomb branch can be determined by means of a secondary product [25].

[8]We will not consider the optional coupling of the bulk gauge fields to a boundary 2d FI parameter in this work.

It is straightforward to see that by shrinking the interval to zero, one obtains the 2d GLSM $\mathcal{T}_{2d}$. This follows from the fact that the 3d vector multiplet is assigned $\mathcal{N}$ at $t = L$, and $\mathcal{D}_m$ at $t = 0$, and so when collapsing the interval the symmetry $T$ is un-gauged and $\varphi$ fixed to the value $m$.

In the following, we will stick for simplicity to Abelian 2d GLSMs $\mathcal{T}_{2d}$, with chiral multiplets $\Phi_s$, $s = 1, \ldots, N$, and Abelian gauge group $G_{2d} = U(1)^N$. We let $Q_s^a$ and $q_s^i$ be the gauge (we mean the 2d gauge group $G_{2d}$ here) and flavour $T$ charges of these chirals, so that the effective mass of $\Phi_s$ is

$$M_s(\sigma, \varphi) = \sum_a Q_s^a \sigma_a + \sum_i q_s^i \varphi_i. \tag{39}$$

## 3.1 Coulomb branch image

We can now describe how one can recover the spectral curve (8) in a natural way from this sandwich representation of the model. Recall that $\mathcal{M}_C = (T^*\mathbb{C}^*)^n$, the bulk Coulomb branch, is precisely the space in which the spectral variety is defined as a holomorphic Lagrangian subvariety. Note also that as the enriched boundary condition $\mathcal{N} \diamond \mathcal{T}_{2d}$ preserves $\mathcal{N} = (2, 2)$ supersymmetry, on general grounds, in the infrared it must also cut out a holomorphic Lagrangian subvariety of $\mathcal{M}_C$, sometimes called the *image* of the boundary condition. We will now show that these two holomorphic Lagrangian subvarieties are the same.

The image of the enriched boundary condition can be computed using the arguments of [8]. Viewing the bulk 3d theory as a 2d $(2, 2)$ theory with an infinite number of fields parameterised by the perpendicular coordinate to the boundary, there is a bulk (twisted) superpotential

$$W_{3d} = \int dt\, \varphi \cdot \partial_t(\sigma_{3d} + i\gamma). \tag{40}$$

In the absence of any boundary matter, the Neumann boundary conditions $\mathcal{N}$ impose (the symbol | here simply means restriction to the right boundary):

$$\sigma_{3d} + i\gamma| \in 2\pi i g^2 \mathbb{Z}^n, \tag{41}$$

or equivalently

$$v_{\pm, i} = 1. \tag{42}$$

We now consider the effect of coupling the bulk theory to $\mathcal{T}_{2d}/T$, our original 2d GLSM with the flavour symmetry gauged by the bulk 3d gauge symmetry. As usual, the presence of the chiral multiplets of $\mathcal{T}_{2d}/T$ induces a 1-loop shift of the effective twisted superpotential [26,27]:

$$\widetilde{W}_{\mathrm{eff}}[\sigma, \varphi|] = -2\pi i \tau \cdot \sigma + \sum_s M_s(\sigma, \varphi|)(\log M_s(\sigma, \varphi|) - 1), \tag{43}$$

where $\tau_a$ is the renormalised 2d FI parameter for the $a^{\mathrm{th}}$ $U(1)$ factor of $G_{2d}$. This is identical to the effective twisted superpotential for the original 2d theory $\mathcal{T}_{2d}$, but with $m$ replaced by the dynamical boundary value of $\varphi$.

The effect of this boundary twisted superpotential is to introduce a delta function contribution to the bulk twisted F-terms (40) which vanish only if the boundary conditions are now deformed to:

$$v_{\mp, i}| = e^{\pm \frac{\partial \widetilde{W}_{\mathrm{eff}}}{\partial \varphi_i}}, \qquad e^{\frac{\partial \widetilde{W}_{\mathrm{eff}}}{\partial \sigma_a}} = 1, \tag{44}$$

that is

$$v_{\mp, i}| = \prod_s (M_s)^{\pm q_s^i}, \qquad \text{for } i = 1, \ldots, n,$$

$$e^{-2\pi i \tau_a} \prod_s (M_s)^{Q_s^a} = 1, \qquad \text{for } a = 1, \ldots, N. \tag{45}$$

The boundary condition for $\varphi$ is deformed in a way to be consistent with the second equation above.[9]

We see that, as expected, these are *precisely* the spectral variety equations discussed in Section 2 and reproduced in particular in (10), where $v_{\mp}$ plays the role of the momenta $p^{\pm 1}$, and $\varphi$ that of $m$. Intuitively, as we will elaborate upon shortly, this is a consequence of the fact that the monopole action at the boundary of the sandwich construction is responsible for shifting the holonomy; the identification (11) therefore tells us that $F_i(w)$ can be obtained by acting with monopole operators at the boundary.

### 3.1.1 Boundary module

The algebra of boundary twisted chiral operators (those in the cohomology of $Q_C| = Q_A$) now form a module for the bulk Coulomb branch algebra. The algebra of boundary local operators are the polynomials in $\sigma, \varphi$ up to the relations imposed by the boundary vacuum relations

$$\mathcal{R}_{2\mathrm{d}} = \mathbb{C}[\sigma_a, \varphi_i]/\{\mathcal{I}\}\,, \tag{46}$$

where $\mathcal{I}$ denotes the ring relations imposed by the 2d vacuum equations $e^{\frac{\partial \widetilde{W}_{\mathrm{eff}}}{\partial \sigma_a}} = 1$. These are precisely the twisted chiral ring relations for the original 2d GLSM, with equivariant parameter $m$ replaced by the dynamical bulk complex scalar $\varphi|$. Thus they reproduce the *quantum equivariant cohomology* $QH^\bullet_T(X)$ of the Higgs branch $X$ of the GLSM. This is a quantum deformation of the (singular) cohomology ring via the contribution of higher degree pseudo-holomorphic curves to correlation functions [28].

The bulk-boundary map of operators (44) show that the boundary local operators $\mathcal{R}_{2\mathrm{d}}$ naturally form a module over the bulk operators $\mathbb{C}[\mathcal{M}_C]$, with support precisely on the spectral variety (8), which is as previously mentioned a holomorphic Lagrangian subvariety. There is a subtlety that is worth mentioning. The bulk monopole operators $v_{\pm,i}$ act at the boundary by multiplication of an element of $\mathcal{R}_{2\mathrm{d}}$ by $e^{\pm \partial \widetilde{W}_{\mathrm{eff}}/\partial \varphi_i}$. Näively, this may seem to take us out of the ring $\mathbb{C}[\sigma_a, \varphi_i]$, since this expression can contain denominators (depending on the effective twisted superpotential). However, physically we expect that since the bulk-boundary map of operators is well-defined, *i.e.* bringing a bulk twisted chiral operator to the boundary results in a boundary twisted chiral operator, and so that this module action is well-defined. Concretely, it means that we expect that after acting with $v_{\pm,i}$, we may re-represent the result as an element of $\mathbb{C}[\sigma_a, \varphi_i]$ by using the ring relations $\mathcal{I}$.[10] Below we show how this works in an example.

### 3.1.2 Example: SQED[2]

For the SQED[2] example considered in Sections 2.1.3 and 2.3.1, it is sufficient to note that

$$v_-| \cdot \mathbf{1} = e^{\frac{\partial \widetilde{W}_{\mathrm{eff}}}{\partial \varphi}} = \frac{\sigma + \varphi|}{\sigma - \varphi|}\,. \tag{47}$$

---

[9]Notice that these equations are in exponential form, which takes into account the Lagrangian multiplier one must add to the action in order to impose the Dirac quantisation conditions for both 3d and 2d gauge fields, once the field strengths are taken to be components of 2d twisted chiral multiplets and unconstrained [17].

[10]More precisely, this is true when $X$ is compact. When $X$ is non-compact, the result may contain denominators in $\varphi_i$. This is due to the fact that the sandwich system, as an effective 2*d* theory, is no longer gapped at certain values of $\varphi$ ($m$ once we sandwich). From the perspective of the Berry connection described in Section 2, it is because *e.g.* the action of $v_{-,i}$ implements the parallel transport $F_i$. For non-compact $X$, the Berry connection is singular at the mass parameters $w$ (or $m$) where the theory is no longer gapped. The parallel transport at these values will thus be singular, and therefore the expansion of the action of $v_{-,i}$ on a basis of the twisted chiral ring will have coefficients rational in $\varphi$. We hope to return to the bulk-boundary description of the case where $X$ is non-compact in future work.

Naïvely, the action has brought us outside of the ring of boundary local operators (since there is a denominator on the RHS of (47)). However, using the chiral ring relations (the same as those presented in (14), but with $m$ replaced by $\varphi$) we get

$$
\begin{aligned}
v_- | \cdot \mathbf{1} = \frac{\sigma + \varphi|}{\sigma - \varphi|} &\sim e^{-2\pi i \tau'}(\sigma + \varphi|)^2 = e^{-2\pi i \tau'}(\sigma^2 + 2\sigma\varphi| + \varphi|^2) \\
&\sim (1 + 2e^{-2\pi i \tau'}\varphi|^2)\mathbf{1} + 2\varphi|e^{-2\pi i \tau'}\sigma \, .
\end{aligned}
\tag{48}
$$

Note that this is consistent with the $\epsilon \to 0$ limit of (33). The action of other bulk operators on other elements of the boundary module can be computed similarly.

## 4 Omega deformation & difference equations

Above we explained how to obtain the Cherkis-Kapustin spectral variety associated to a Berry connection of a 2d GLSM, which was defined in Section 2.1 using the $tt^*$ geometry methods uncovered in [3,4], by considering instead a 3d sandwich representation of the model. The advantage of this viewpoint is that it makes the connection to the action of 3d $\mathcal{N} = 4$ Coulomb branch chiral ring on the quantum equivariant cohomology of the target of the 2d GLSM particularly manifest. The aim of this section is to explain how the difference equations for vortex partition functions reviewed in Section 2.3 can be recovered from a closely related setup. Whether this can be done for the more general difference equations presented in Section 2.2 is an interesting question that we leave to future work.

### 4.1 Setup

To derive the difference equations, we introduce in the previous setup an Omega background. This is a deformation of the Lagrangian and in particular of the supercharge $Q_C$ (whose cohomology ring we recall corresponds to the 3d $\mathcal{N} = 4$ Coulomb branch operators), such that

$$
Q_C^2 = \epsilon \mathcal{L}_V \, ,
\tag{49}
$$

where $V$ generates rotations about the origin in $\mathbb{R}^2$. At the boundary $Q_C| = Q_A$ restricts to the 2d A-model supercharge, and thus this bulk Omega deformation reduces to the 2d Omega deformation at the boundary. We will denote the space-time $I \times \mathbb{R}^2$ with the above Omega deformation inserted along $\mathbb{R}^2$ by $I \times \mathbb{R}^2_\epsilon$, or $\mathbb{R}^+ \times \mathbb{R}_\epsilon$ when we consider the same setup on a half-space.

The Omega deformation effectively reduces the system to a quantum mechanics along the fixed axis of rotation, and the bulk Coulomb branch operators must be inserted along this axis. The algebra of bulk operators is therefore deformed to a non-commutative operator algebra $\hat{\mathbb{C}}_\epsilon[\mathcal{M}_C]$ which reduces to $\mathbb{C}[\mathcal{M}_C]$ as $\epsilon \to 0$. For the pure Abelian gauge theory we consider here, this is specified by the relations

$$
[\hat{\varphi}_i, \hat{v}_{\pm,j}] = \pm \epsilon \delta_{ij} \hat{v}_{\pm,i} \, , \qquad \hat{v}_{+,i} \hat{v}_{-,i} = 1 \, .
\tag{50}
$$

Clearly, in the presence of the Omega deformation, the previous notion of boundary twisted chiral ring (which was found to be isomorphic to the quantum equivariant cohomology ring of the target of the GLSM) ceases to exist. However, the bulk quantised Coulomb branch algebra still acts on the space of 2d twisted chiral operators on the boundary supporting the gauged 2d GLSM $\mathcal{T}_{2d}/T$. Thus, this boundary still defines a certain module for the quantised algebra. This module can be studied along the lines of the modules described in [8]. More mathematically, they should be related to constructions proposed in Section 7 of [9]. In this article, we analyse these modules with the purpose of recovering the difference equations for vortex partition functions of Section 2.3.

### 4.1.1 Right boundary module

For simplicity, we restrict the following discussion to $n = 1$. For general $n$, one simply needs to introduce appropriate (and obvious) indices. Let us first consider the system on $\mathbb{R}^+ \times \mathbb{R}_\epsilon$, with the enriched Neumann boundary condition $\mathcal{N} \diamond \mathcal{T}_{2d}$. To be a little more precise, notice that for the effective quantum mechanics resulting from the Omega deformation to make sense, we also need to impose a boundary condition at infinity of $\mathbb{R}^2_\epsilon$. We can choose this to be a fixed vacuum $\alpha$.[11] Then the boundary condition $\mathcal{N} \diamond \mathcal{T}_{2d}$ we will generate a state in the supersymmetric quantum mechanics, which we can denote by

$$|\mathcal{N} \diamond \mathcal{T}_{2d}, \alpha\rangle . \tag{51}$$

Specifically, there should be as in [21] a canonical map of modules

$$l : M \to \mathcal{H}_\alpha , \tag{52}$$

where $\mathcal{H}_\alpha$ is the Hilbert space of the theory on $\mathbb{R}^2_\epsilon$ where the fields are required to lie in the vacuum $\alpha$ at spatial infinity, and $M$ is the module of boundary local (twisted chiral) operators supported by the enriched boundary condition $\mathcal{N} \diamond \mathcal{T}_{2d}$. The map is simply given by the insertion of the respective local operator $\mathcal{O}^{2d}$ at the boundary. It is canonical because the identity insertion simply corresponds to the state generated by the boundary condition:

$$l(\mathbf{1}) = |\mathcal{N} \diamond \mathcal{T}_{2d}, \alpha\rangle . \tag{53}$$

We will more generally denote the image of $l(\mathcal{O}^{2d})$ in $\mathcal{H}_\alpha$ by

$$l(\mathcal{O}^{2d}) := |\mathcal{N} \diamond \mathcal{T}_{2d}, \alpha, \mathcal{O}^{2d}\rangle , \tag{54}$$

so that in particular

$$|\mathcal{N} \diamond \mathcal{T}_{2d}, \alpha, \mathbf{1}\rangle = |\mathcal{N} \diamond \mathcal{T}_{2d}, \alpha\rangle . \tag{55}$$

Consider now a $\mathbb{C}[\varphi]$ basis for the module of boundary local operators; which is simply a basis for the twisted chiral ring of the original 2d theory $\mathcal{T}_{2d}$ (since we obtain this from the local operators supported at the boundary by setting $\varphi$ to be a constant). This will consist of polynomials (or symmetric polynomials in the non-Abelian case) in the complex scalars $\sigma$. We denote such a basis by $\{\mathcal{O}_a\}$ as before.

We can consider the action of bulk Coulomb branch operators on such a basis, by bringing them to act on the boundary and re-expanding. In particular, we claim that in $M$:

$$\hat{v}_- \cdot \mathcal{O}_a = \sum_b \widetilde{G}(\varphi, \epsilon)_{ab} \mathcal{O}_b , \tag{56}$$

where $\mathcal{O}_b$ are other twisted chiral ring operators of the original $\mathcal{T}_{2d}$, i.e. a polynomial in $\sigma$. $\widetilde{G}(\varphi, \epsilon)$ encodes the coefficients of this re-expansion. We demonstrate how this works concretely for Abelian GLSMs momentarily. Importantly, the map $l$ commutes by construction with the action of $\hat{\mathbb{C}}_\epsilon[\mathcal{M}_C]$, and so we get

$$\hat{v}_- |\mathcal{N} \diamond \mathcal{T}_{2d}, \alpha, \mathcal{O}_a\rangle = \hat{v}_- l(\mathcal{O}_a) = l(\hat{v}_- \cdot \mathcal{O}_a) = \sum_b \widetilde{G}(\varphi, \epsilon)_{ab} |\mathcal{N} \diamond \mathcal{T}_{2d}, \alpha, \mathcal{O}_b\rangle . \tag{57}$$

This equation, as well as a similar equation for $\hat{v}_+$, defines the image of $M$ (strictly the basis $\mathcal{O}_a$, but the extension to the entirety of $M$ is trivial) under $l$ as a module for the quantised Coulomb branch algebra. We will demonstrate how to compute the matrix $\widetilde{G}(\varphi, \epsilon)_{ab}$ in Section 4.2, but first we will show how to derive from the above difference equations for vortex partition functions, which will be identified with those studied in Section 2.3.

---

[11]Alternatively, we could take the setup to be $I \times HS^2$ and place a finite boundary condition at $\partial HS^2$. To make the analogy with [8] we work with the former setup for now, but the constructions also all hold for $\mathbb{R}^2$ compactified to $HS^2$ with a boundary condition at finite distance, and replacing 'vortex partition function' with 'hemisphere partition function' throughout this article. More comments will be provided in Section 4.4

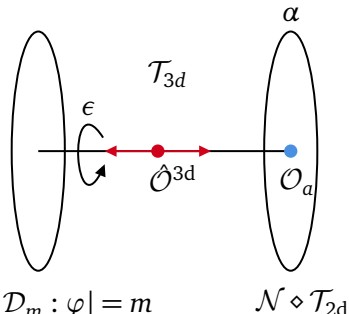

Figure 2: The sandwich construction for difference modules.

### 4.1.2 Left boundary module & vortex partition function

A vortex partition function for the 2d GLSM $\mathcal{T}_{2d}$ can be defined in terms of the overlap,

$$\mathcal{Z}_\alpha[\mathcal{O}_a, m] := \langle \mathcal{D}_m | \mathcal{N} \diamond \mathcal{T}_{2d}, \mathcal{O}_a, \alpha \rangle \,, \tag{58}$$

between states in the effective supersymmetric quantum mechanics on a compact interval $I$. Here $|\mathcal{N} \diamond \mathcal{T}_{2d}, \mathcal{O}_a, \alpha\rangle$ are the states considered in (54), whereas $\langle \mathcal{D}_m |$ are states generated by a Dirichlet boundary conditions where

$$|\varphi = m \,. \tag{59}$$

This is because, as we mentioned before, the length of the interval is $Q_C$-exact, so we can shrink it to zero. In the absence of bulk operators in the interval, since the 3d vector multiplet is assigned $\mathcal{N}$ at $t = 0$, and $\mathcal{D}_m$ at $t = L$, there are no resulting 2d degrees of freedom after collapsing the interval, and $\varphi$ is fixed to the value $m$. If we have also inserted a 2d twisted chiral ring operator $\mathcal{O}_a$ at the origin at the boundary, after collapsing we obtain precisely the 2d theory $\mathcal{T}_{2d}$ with equivariant parameter (complex mass) $m$, with this insertion. The 3d path integral on $I \times \mathbb{R}_\epsilon$ therefore computes the vortex partition function of $\mathcal{T}_{2d}$ for vacuum $\alpha$.

In order to derive the difference equation (25), let us then consider the insertion of elements of the quantised Coulomb branch algebra in the bulk. The setup we want is as illustrated in Figure 2.
The path integral then computes

$$\langle \mathcal{D}_m | \hat{O}^{3d} | \mathcal{N} \diamond \mathcal{T}_{2d}, \mathcal{O}_a, \alpha \rangle \,. \tag{60}$$

We claim that the difference equations can be derived by bringing the bulk monopole operator to act on either the top or the right boundary before collapsing the interval.

It is not difficult to see that if we act on the state generated by the left boundary, then the action simply shifts the boundary value of $|\varphi$ to $m + \epsilon$

$$\langle \mathcal{D}_m | \hat{O}^{3d} = \langle \mathcal{D}_{m+\epsilon} | \,. \tag{61}$$

This follows from the fact that the action of the operators $\hat{v}_\pm$ and $\varphi$ of the supersymmetric quantum mechanics along the interval is the same on both sides. Thus, if we denote the operator shifting $m$ by $+\epsilon$ by $\hat{p}$ we obtain, for $\hat{O}^{3d} = \hat{v}_-$:

$$\langle \mathcal{D}_m | \hat{v}_- | \mathcal{N} \diamond \mathcal{T}_{2d}, \alpha, \mathcal{O}_a \rangle = \hat{p} \, \mathcal{Z}_\alpha[\mathcal{O}_a, m] \,. \tag{62}$$

On the other hand, it follows from (56) that

$$
\begin{aligned}
\langle \mathcal{D}_m | \hat{v}_- | \mathcal{N} \diamond \mathcal{T}_{2d}, \alpha, \mathcal{O}_a \rangle &= \langle \mathcal{D}_m | \mathcal{N} \diamond \mathcal{T}_{2d}, \alpha, \sum_b \widetilde{G}(\varphi, \epsilon)_{ab} \mathcal{O}_b \rangle \\
&= \sum_b \widetilde{G}(m, \epsilon)_{ab} \langle \mathcal{D}_m | \mathcal{N} \diamond \mathcal{T}_{2d}, \alpha, \mathcal{O}_b \rangle \\
&= \sum_b \widetilde{G}(m, \epsilon)_{ab} \, \mathcal{Z}_\alpha[\mathcal{O}_b, m].
\end{aligned}
\tag{63}
$$

Combining equations (62) and (63) we obtain exactly the first-order difference equations of the desired form

$$
\hat{p} \, \mathcal{Z}_\alpha[\mathcal{O}_a, m] = \sum_b \widetilde{G}(m, \epsilon)_{ab} \, \mathcal{Z}_\alpha[\mathcal{O}_b, m].
\tag{64}
$$

We will shortly show that these equations precisely coincide with (25).[12] Of course, insertions of $\mathbb{C}[\hat{\varphi}]$ will result in a constant multiplication by an element of $\mathbb{C}[m]$ after sandwiching. Further, insertions of $\hat{v}_+$ will result in a first-order difference equation for $\hat{p}^{-1}$, which by consistency with the bulk Coulomb branch algebra relations must be

$$
\hat{p}^{-1} \mathcal{Z}_\alpha[\mathcal{O}_a, m] = \sum_b \widetilde{G}(m - \epsilon, \epsilon)_{ab}^{-1} \mathcal{Z}_\alpha[\mathcal{O}_b, m].
\tag{65}
$$

## 4.2 Enriched module of boundary local operators

We have seen that to determine the equations that are obeyed by the vortex partition functions, one can just determine the action of the monopole operators on the module (56). We now explain how this action can be computed explicitly via localisation methods.

### 4.2.1 Higgs branch reminder

For inspiration, let us recall what happens when the bulk theory is instead a 3d free matter theory (the Higgs branch case), see Section 5.2 of [8]. For a free hypermultiplet, we simply have $[\hat{X}, \hat{Y}] = \epsilon$, in the usual Omega background. Suppose we place Neumann conditions on $X$ and Dirichlet on $Y$. Then the boundary module consists of polynomials $\mathbb{C}[X]$, and the bulk algebra acts as:

$$
\hat{X} : \times X, \qquad \hat{Y} = \epsilon \partial_X.
\tag{66}
$$

If one couples this bulk to a boundary 2d chiral $\phi$, via a superpotential $W(X, \phi)$, we have that:

- The boundary module is enlarged to $\mathbb{C}[X, \phi]$.

- The bulk algebra action is conjugated by $e^{\frac{W}{\epsilon}}$, so

$$
\hat{X} : X \times, \qquad \hat{Y} = \epsilon \partial_X + \partial_X W.
\tag{67}
$$

---

[12]It is worth commenting at this point on the relationship between the above construction and [21], where a similar setup was used to derive differential (Picard-Fuchs) equations for the vortex partition functions. There, Neumann boundary conditions for a bulk 3d gauge theory with gauge group $G$ with hypermultiplets were instead imposed on both ends of an interval on $I \times \mathbb{R}^2$. The boundary conditions for hypermultiplets are specified by Lagrangian splittings $L$, $L'$, (in $\mathcal{N} = 2$ language specifying Dirichlet to one $\mathcal{N} = 2$ chiral multiplet in each hypermultiplet, and Neumann to the other). Collapsing the interval gives the vortex partition function for a 2d GLSM with gauge group $G$ and matter $L \cap L'$. Inserting bulk monopole operators in the interval along the axis of the Omega deformation and considering their action on the space of boundary local operators leads to *differential* equations for vortex partition functions. The setup we consider here is slightly different; the degrees of freedom of the 2d theory of interest are not split between two boundaries and a bulk theory, but lie at a single boundary. The flavour symmetry is gauged by a theory in the bulk, with equivariant parameters fixed by a second, Dirichlet boundary.

- Boundary operators in $\mathrm{Im}(\epsilon\partial_\phi + \partial_\phi W)$ are set to 0.

The origin of the latter two points is that for a 2d LG model in the Omega background, the expectation value of a chiral operator at the boundary is given by the integral

$$\langle \mathcal{O}(\phi)\rangle = \int_\gamma e^{\frac{W}{\epsilon}}\mathcal{O}(\phi)\Omega, \tag{68}$$

where $\Omega$ is the holomorphic top form on the target space and $\gamma$ is a middle-dimensional Lagrangian that encodes the boundary conditions at infinity. Therefore, operators in $\mathrm{Im}(\epsilon\partial_\phi + \partial_\phi W)$ become exact derivatives and set to 0, and further the action of bulk operators should be conjugated by the (exponential) of the superpotential to act properly inside correlators.

### 4.2.2 Coulomb branch module

We now propose a similar recipe for the Coulomb branch setup discussed in this article. For a standard (not enriched) Neumann boundary condition, the boundary module consists simply of the polynomials $\mathbb{C}[\varphi]$ in the complex scalar, and we have that:

$$\hat{v}_\pm \cdot f(\varphi) = f(\varphi \pm \epsilon), \qquad \hat{\varphi} f(\varphi) = \varphi f(\varphi). \tag{69}$$

This is complicated by the presence of additional matter and interactions on the boundary.

Suppose we now couple to the boundary theory $\mathcal{T}_{2d}$ (by gauging its flavour symmetry). As we explained above, for a 2d theory in Omega background, the expectation values of twisted chiral operators are computed as the vortex (or hemisphere) partition functions. For the GLSMs considered here, and with $r$ denoting the rank of the gauge group, these can schematically be computed as follows

$$\mathcal{Z}[\mathcal{O}(\sigma), m] = \oint_\alpha \frac{d^r\sigma}{(2\pi i\epsilon)^r}\mathcal{I}[\sigma, m, \tau, \epsilon]\mathcal{O}[\sigma]. \tag{70}$$

Here $\mathcal{I}[\sigma, m, \tau, \epsilon]$ is the integrand of the vortex partition function (obtained via Coulomb branch localisation), and $r$ is the rank of the gauge group.

Similarly to the Higgs branch case, we can then propose the following recipe:

- Enlarge the module of boundary local operators to $\mathbb{C}[\sigma, \varphi]$.

- Conjugate the action of bulk operators by $\mathcal{I}$, *i.e.*

$$\hat{v}_{-,i} = \mathcal{I}^{-1}\hat{p}_i\mathcal{I}, \qquad \hat{v}_{+,i} = \mathcal{I}^{-1}\hat{p}_i^{-1}\mathcal{I}, \qquad \hat{\varphi} = \times\varphi. \tag{71}$$

Notice that the conjugation preserves the bulk Coulomb branch algebra relations.

- Quotient by the ideal of polynomials for which the expectation values in (70) vanish. That is, quotient by the polynomials:

$$\mathrm{Im}(\mathcal{I}^{-1}\hat{P}_a\mathcal{I} - 1), \tag{72}$$

where $\hat{P}_a : \sigma_a \mapsto \sigma_a + \epsilon$. This is because:

$$\left\langle \left(\mathcal{I}^{-1}\hat{P}_a\mathcal{I} - 1\right)f[\sigma]\right\rangle = \oint_\alpha \frac{d\sigma}{2\pi i\epsilon}\left(\hat{P}_a - 1\right)\mathcal{I}f, \tag{73}$$

which is a total derivative as $\hat{P}_a - 1 = \epsilon\partial_\sigma + \frac{\epsilon^2}{2}\partial_\sigma^2 + \ldots$

We explore what this means for Abelian GLSMs below.

## 4.3 Abelian GLSMs

In the case of an Abelian GLSM, the integrand in the vortex partition function is given by:

$$
\mathcal{I}[\sigma, m, \tau, \epsilon] = e^{-\frac{2\pi i \tau \cdot \sigma}{\epsilon}} \sum_s \Gamma\left[\frac{M_s(\sigma, m)}{\epsilon}\right].
$$
(74)

The previous recipe can be specialised to:

1. The module is first enlarged to $\mathbb{C}[\sigma, \varphi]$.

2. The monopole operators are modified to act on this space as:

$$
\hat{v}_{-,i} = \frac{\prod_{s|q_s^i>0}\left[\frac{M_s}{\epsilon}\right]_{q_s^i}}{\prod_{s|q_s^i<0}\left[\frac{M_s}{\epsilon}\right]_{q_s^i}} \hat{p}_i,
$$

$$
\hat{v}_{+,i} = \frac{\prod_{s|q_s^i<0}\left[\frac{M_s}{\epsilon}\right]_{-q_s^i}}{\prod_{s|q_s^i>0}\left[\frac{M_s}{\epsilon}\right]_{-q_s^i}} \hat{p}_i^{-1}.
$$
(75)

Here the quantum product is given by

$$
[a]_k = \begin{cases} \prod_{i=0}^{k-1}(a+i), & \text{if } k > 0, \\ \prod_{i=1}^{|k|}(a-i), & \text{if } k < 0, \end{cases}
$$
(76)

and the shift operator remains $\hat{p}_i = e^{\epsilon \partial_{m_i}}$.

3. We quotient by the polynomials in the image of:

$$
e^{-2\pi i \tau_a} \frac{\prod_{s|Q_s^a>0}\left[\frac{M_s}{\epsilon}\right]_{Q_s^a}}{\prod_{s|Q_s^a<0}\left[\frac{M_s}{\epsilon}\right]_{Q_s^a}} \hat{P}_a - 1.
$$
(77)

That is, we identify polynomials:

$$
\left(e^{-2\pi i \tau_a} \frac{\prod_{s|Q_s^a>0}\left[\frac{M_s}{\epsilon}\right]_{Q_s^a}}{\prod_{s|Q_s^a<0}\left[\frac{M_s}{\epsilon}\right]_{Q_s^a}} \hat{P}_a\right) f(\sigma, \varphi) \sim f(\sigma, \varphi),
$$
(78)

where $\hat{P}_a : \sigma_a \mapsto \sigma_a + \epsilon$ is the shift operator on $\sigma$.

It is not hard to check that the above is consistent with the Coulomb branch algebra (50).

In practice, we shall see that acting with (75) näively takes us out of $\mathbb{C}[\sigma, \varphi]$. However, using the identifications (78) always enables us to identify the result with an element of $\mathbb{C}[\sigma, \varphi]$. This is the analogue of the statements made in Section 3.1.1 and indeed, the quotient (78) recovers the quotient by the vacuum ideal (46) in the $\epsilon \to 0$ limit.

## 4.4 Hemisphere partition functions

We note how the arguments above are the same for the hemisphere partition functions, which arise in the setup on $I \times HS^2$ instead of $I \times \mathbb{R}_\epsilon^2$, where the radius of the hemisphere is identified as $\epsilon^{-1}$. The chiral multiplets of $\mathcal{T}_{2d}$ can be given Neumann or Dirichlet boundary conditions

on $\partial HS^2 \cong S^1$. Doing so for the $s^{\text{th}}$ chiral multiplet results in a contribution to the integrand for the partition function (to be integrated over Coulomb branch scalars):

$$(N): \quad \mathcal{I}_N\left[\frac{M_s}{\epsilon}\right] = \Gamma\left[\frac{M_s(\sigma, m)}{\epsilon}\right], \qquad (D): \quad \mathcal{I}_D\left[\frac{M_s}{\epsilon}\right] = \frac{-2\pi i e^{\pi i \frac{M_s(\sigma, m)}{\epsilon}}}{\Gamma\left[1 - \frac{M_s(\sigma, m)}{\epsilon}\right]}. \tag{79}$$

Thus, assigning Neumann boundary conditions results in an integrand identical to (74), and thus the vortex partition function.

Taking (70) to be an expectation value in the more general hemisphere partition function, and $\mathcal{I}$ to be the integrand for the hemisphere partition function (now allowing Dirichlet boundary conditions for chirals), results in the same action on the boundary twisted chiral ring. That is, equations (56)-(78) are unchanged. This is because the result of conjugating the shift operators by either of (79) is the same. This follows because of the identities:

$$\frac{\mathcal{I}_D[x+n]}{\mathcal{I}_D[x]} = \frac{\mathcal{I}_N[x+n]}{\mathcal{I}_N[x]} = \begin{cases} [x]_n, & \text{if } n > 0, \\ \frac{1}{[x]_{-n}}, & \text{if } n < 0. \end{cases} \tag{80}$$

## 4.5 Example: SQED[N]

To test our physical recipe we now discuss the theory $\mathcal{T}_{2d}$ SQED[N], defined in a similar way to SQED[2] but with $N$ chiral multiplets. We introduce masses $m_1, \ldots, m_{N-1}$ for the $T = U(1)^{N-1}$ flavour symmetry on $N$ chiral multiplets, which have effective complex masses:

$$M_i[\sigma, m] = \sigma + m_i, \qquad i = 1, \ldots, N, \tag{81}$$

where we identify always

$$m_N = -\sum_{j=1}^{N-1} m_j. \tag{82}$$

Gauging the symmetry, we have $m_i \mapsto \varphi_i$, $i = 1, \ldots, N-1$ complex scalars of a bulk 3d gauge theory $\mathcal{T}_{3d}$ with gauge group $T = U(1)^{N-1}$. The boundary module consists of elements of $\mathbb{C}[\sigma, \varphi]$ quotiented by the submodule of elements in the image of:

$$q \prod_{j=1}^{N} (\sigma + \varphi_j) \hat{P} - 1, \tag{83}$$

where $\hat{P} = e^{\epsilon \partial_\sigma}$ and $q := \epsilon^{-N} e^{-2\pi i \tau}$. The bulk Coulomb branch operators are quantised to:

$$\hat{v}_{-,i} = \frac{\sigma + \varphi_i}{\sigma + \varphi_N - \epsilon} \hat{p}_i, \qquad \hat{v}_{+,i} = \frac{\sigma + \varphi_N}{\sigma + \varphi_i - \epsilon} \hat{p}_i^{-1}, \tag{84}$$

where $\varphi_N = -\varphi_1 - \ldots - \varphi_{N-1}$ so that $\hat{p}_i : \varphi_N \mapsto \varphi_N - \epsilon$.

For a general element $f(\sigma, \varphi)$ in $\mathbb{C}[\sigma, \varphi]$, we have that:

$$\begin{aligned} \hat{v}_{-,i} f(\sigma, \varphi) &= \frac{\sigma + \varphi_i}{\sigma + \varphi_N - \epsilon} f(\sigma, \varphi_i + \epsilon) \\ &\sim q \prod_{j=1}^{N} (\sigma + \varphi_j) \hat{P} \frac{\sigma + \varphi_i}{\sigma + \varphi_N - \epsilon} f(\sigma, \varphi_i + \epsilon) \\ &= q \prod_{j=1}^{N} (\sigma + \varphi_j) \frac{\sigma + \varphi_i + \epsilon}{\sigma + \varphi_N} f(\sigma + \epsilon, \varphi_i + \epsilon) \\ &= q(\sigma + \varphi_i + \epsilon) \left( \prod_{j=1}^{N-1} (\sigma + \varphi_j) \right) f(\sigma + \epsilon, \varphi_i + \epsilon), \end{aligned} \tag{85}$$

where $\sim$ indicates we have used the identification (83). We also have that

$$
\begin{aligned}
\hat{v}_{+,i} f(\sigma, \varphi) &= \frac{\sigma + \varphi_N}{\sigma + \varphi_i - \epsilon} f(\sigma, \varphi_i - \epsilon) \\
&\sim q \prod_{j=1}^{N} (\sigma + \varphi_j) \hat{P} \frac{\sigma + \varphi_N}{\sigma + \varphi_i - \epsilon} f(\sigma, \varphi_i - \epsilon) \\
&= q(\sigma + \varphi_N + \epsilon) \left( \prod_{i \neq j}^{N} (\sigma + \varphi_j) \right) f(\sigma + \epsilon, \varphi_i - \epsilon).
\end{aligned}
\tag{86}
$$

It is also not hard to check that

$$
\begin{aligned}
\hat{v}_{+,i} \hat{v}_{-,i} f(\sigma, \varphi) &= q^2 \left( \prod_{j=1}^{N} (\sigma + \varphi_j)(\sigma + \varphi_j + \epsilon) \right) f(\sigma + 2\epsilon, \varphi) \\
&= \left( q \prod_{j=1}^{N} (\sigma + \varphi_j) \hat{P} \right)^2 f(\sigma, \varphi) \\
&\sim f(\sigma, \varphi),
\end{aligned}
\tag{87}
$$

and similarly for $\hat{v}_{-,i} \hat{v}_{+,i}$. The quantisations are therefore consistent with the Coulomb branch algebra relations.

Note also that (83) imply that any element of the boundary module is equivalent to an element of $\mathbb{C}[\sigma, \varphi]$ with powers of $\sigma$ at most $N - 1$. That is, $\{1, \sigma, \ldots, \sigma^{N-1}\}$ remains a $\mathbb{C}[\varphi]$ basis of the module (and once we sandwich with the Dirichlet, these coefficients are fixed to be constant anyway). Concretely, (83) implies for $k \geq N$ that:

$$
q \prod_{j=1}^{N} (\sigma + \varphi_j) \hat{P} (\sigma - \epsilon)^{k-N} \sim (\sigma - \epsilon)^{k-N},
\tag{88}
$$

and so

$$
\sigma^k \sim \left( \prod_{j=1}^{N} (\sigma + \varphi_j) - \sigma^N \right) \sigma^{k-N} + q^{-1} (\sigma - \epsilon)^{k-N}.
\tag{89}
$$

The left-hand side is a polynomial of order $k - 1$. This can be used repeatedly to reduce any element of the boundary module to a representative with maximum degree $N - 1$ in $\sigma$.

### 4.5.1  Example: SQED[2]

Let us test the above results in the example of SQED[2] and check consistency with the results obtained in Section 2.1.3 and 2.3.1. We will further specify the results to SQED[3] in Appenidx A, and check consistency with the results of [3].

For SQED[2] we have

$$
\hat{v}_- = \frac{\sigma + \varphi}{\sigma - \varphi - \epsilon} \hat{p},
\tag{90}
$$

and should identify in $\mathbb{C}[\sigma, \varphi]$

$$
\frac{e^{-2\pi i \tau}}{\epsilon^2} (\sigma + m)(\sigma - m) f(\sigma + \epsilon) \sim f(\sigma).
\tag{91}
$$

Let us consider the action on $\mathcal{O}_a \in \{\mathbf{1}, \sigma\}$

$$
\begin{aligned}
\hat{v}_- \cdot \mathbf{1} &= \frac{\sigma + \varphi}{\sigma - \varphi - \epsilon} \\
&= 1 + \frac{2\varphi + \epsilon}{\sigma - \varphi - \epsilon} \\
&\sim 1 + (2\varphi + \epsilon)(\sigma + \varphi)e^{-2\pi i\tau} \\
&= (1 + (2\varphi + \epsilon)\varphi\, e^{-2\pi i\tau})\mathbf{1} + (2\varphi + \epsilon)e^{-2\pi i\tau}\sigma \,.
\end{aligned}
\tag{92}
$$

Notice how these coefficients of $\mathcal{O}_a$ match the first row of (33). Similarly:

$$
\begin{aligned}
\hat{v}_- \cdot \sigma &= \frac{(\sigma + \varphi)\sigma}{\sigma - \varphi - \epsilon} \\
&= \sigma + (2\varphi + \epsilon) + \frac{(2\varphi + \epsilon)(\varphi + \epsilon)}{\sigma - \varphi - \epsilon} \\
&\sim (2m + \epsilon)(1 + (m + \epsilon)m\, e^{-2\pi i\tau})\mathbf{1} + (1 + e^{-2\pi i\tau}(\varphi + \epsilon)(2\varphi + \epsilon))\sigma \,,
\end{aligned}
\tag{93}
$$

matching the second row of (33).

# 5 Bulk gauge theories with matter

Above we have discussed an action of the (quantised) Coulomb branch algebra of a pure Abelian gauge theory with gauge group $T$ on $QH_T^\bullet(X)$, where $X$ is the vacuum manifold of the GLSM. It is natural to ask whether we can bulk theories more general than pure gauge theory, for example a 3d Abelian gauge theory with matter in some representation. Mathematically, for a 3d gauge theory with gauge group $T$ and hypermultiplets transforming in a representation $L \oplus L^*$, where $L \cong \mathbb{C}^N$ specifies a polarisation, one would expect to find an action of the Coulomb branch algebra on $QH_T^\bullet(X \times L)$ (appropriately defined, as $L$ is non-compact), and the Omega-deformed version of this [9]. For the purposes of this article, we show that this is a mild modification of the above constructions, and that the sandwich results in the same difference equations as before.

Let us first take $\mathcal{T}_{2d}$ to be trivial, and therefore consider an Abelian $\mathcal{T}_{3d}$ with gauge group $T$ and matter $L \oplus L^*$ with Neumann boundary conditions at one end of the sandwich. The $(2,2)$ boundary conditions dictate that one sets the chiral multiplets in $L^*$ to zero at the boundary (Dirichlet), and allows those in $L$ to fluctuate (Neumann). For an Abelian theory one can think of $L$ as a vector of $\tilde{N}$ signs, where $L_\alpha = \pm$ if $X_\alpha$ is Neumann/Dirichlet (and the opposite for $Y_\alpha$). Different boundary conditions may be obtained by decomposing the hypermultiplet representation by different Lagrangian splittings.

We denote the Coulomb branch by $\mathcal{M}_C(T, L \oplus L^*)$. The bulk Coulomb branch algebra is [24]:

$$
v_A v_B = v_{A+B} P_{A,B}(\varphi), \qquad P_{A,B}(\varphi) = \prod_{\alpha=1}^{\tilde{N}} (\tilde{M}_\alpha)^{(\tilde{Q}_\alpha^A)_+ + (\tilde{Q}_\alpha^B)_+ - (\tilde{Q}_\alpha^{A+B})_+} \,,
\tag{94}
$$

where effective complex masses are $\tilde{M}_\alpha = \tilde{Q}_\alpha^i \varphi_i$. Here $\tilde{Q}_\alpha^A$ is the charge of $X_\alpha$ under charge generator $A \in \mathrm{Hom}(U(1), T)$ and $(x)_+ = \max(x, 0)$.

The boundary twisted chiral ring is given by polynomials $\mathbb{C}[\varphi]$, which can be identified with $QH_T^\bullet(L)$. The boundary conditions specify (for a right boundary condition) that

$$
v_A| = \prod_{\alpha=1}^{\tilde{N}} (\tilde{M}_{\alpha,L})^{(\tilde{Q}_{A,L}^\alpha)_+} \,,
\tag{95}
$$

which specifies the action of the bulk Coulomb branch algebra on the boundary chiral ring, i.e. $\mathbb{C}[\mathcal{M}_C(T, L \oplus L^*)]$ on $QH_T^\bullet(L)$. Here $\tilde{M}_{\alpha,L} = L_\alpha \tilde{M}_\alpha$.

Now, suppose we couple $\mathcal{T}_{3d}$ to a 2d theory $\mathcal{T}_{2d}$ by gauging the flavour symmetry $T$ of the 2d theory. There is an induced 1-loop correction to the boundary twisted superpotential (43). This deforms the boundary conditions (95) to:

$$
\begin{aligned}
\nu_A| &= \prod_{\alpha=1}^{\tilde{N}} (\tilde{M}_{\alpha,L})^{(\tilde{Q}_{\alpha,L}^A)_+} e^{-A_i \frac{\partial \widetilde{W}_{\text{eff}}}{\partial \varphi_i}} \\
&= \prod_{\alpha=1}^{\tilde{N}} (\tilde{M}_{\alpha,L})^{(\tilde{Q}_{\alpha,L}^A)_+} \prod_s M_s^{-q_s^A},
\end{aligned}
\tag{96}
$$

where $q_s^A = \sum_i q_s^i A_i$ is the charge of the $s^{\text{th}}$ chiral multiplet under charge generator $A$. Note this deformation is compatible with the Coulomb branch algebra. The boundary vacuum equations simultaneously impose:

$$
e^{-2\pi i \tau_a} \prod_s (M_s)^{Q_s^a} = 1,
\tag{97}
$$

as before. Together these equations cut out a holomorphic Lagrangian submanifold in $\mathcal{M}_C(T, L \oplus L^*)$.

The boundary twisted chiral ring of operators is generated by $\sigma_a$ and $\varphi_i$, with the standard chiral ring relations imposed. In this setup, it is natural to identify this ring with (some version of) $QH_T^\bullet(X \times L)$.[13] Thus, the equations (96) exhibit the elements of $QH_T^\bullet(X \times L)$ as a module of the bulk Coulomb branch algebra, with support on the holomorphic Lagrangian submanifold of $\mathcal{M}_C(T, L \oplus L^*)$ cut out by the simultaneous equations (96) and (97).

## 5.1 Omega deformation

In the presence of the Omega deformation, similar to the pure gauge theory case, there is no longer a notion of boundary twisted chiral ring. However, by the same arguments put forward in Section 3, boundary twisted chiral operators still form a module over the quantised bulk Coulomb branch algebra. We now explain in more detail how this works, and how to obtain the difference equations.

Firstly, the bulk Coulomb branch algebra is quantised to $\hat{\mathbb{C}}_\epsilon[\mathcal{M}_C(T, L \oplus L^*)]$:[14]

$$
[\hat{\varphi}_i, \hat{v}_A] = \epsilon A_i \hat{v}_A, \qquad \hat{v}_A \hat{v}_B = P_{A,B}^l(\hat{\varphi}) \hat{v}_{A+B} P_{A,B}^r(\hat{\varphi}),
\tag{98}
$$

where

$$
P_{A,B}^l(\hat{\varphi}) = \prod_{\substack{\alpha \mid |\tilde{Q}_\alpha^A| \le |\tilde{Q}_\alpha^B| \\ \tilde{Q}_\alpha^A \tilde{Q}_\alpha^B < 0}} \epsilon^{|\tilde{Q}_\alpha^A|} \left[ \frac{\hat{M}_\alpha + \frac{\epsilon}{2}}{\epsilon} \right]^{-\tilde{Q}_\alpha^A}, \qquad P_{A,B}^r(\hat{\varphi}) = \prod_{\substack{\alpha \mid |\tilde{Q}_\alpha^A| > |\tilde{Q}_\alpha^B| \\ \tilde{Q}_\alpha^A \tilde{Q}_\alpha^B < 0}} \epsilon^{|\tilde{Q}_\alpha^B|} \left[ \frac{\hat{M}_\alpha + \frac{\epsilon}{2}}{\epsilon} \right]^{\tilde{Q}_\alpha^B}.
\tag{99}
$$

The shift by $\frac{\epsilon}{2}$ is due to the $R_V$-charge $1/2$ of the bulk chiral multiplets.

---

[13]Notice that the ring we presented above for $QH_T^\bullet(X \times L)$ turns out to be isomorphic to $QH_T^\bullet(X)$. This is consistent with the fact that the quantum *equivariant* cohomology of an affine space $L$ is simply isomorphic to polynomials in the equivariant parameter. However, the presence of $\mathcal{N} = (2,2)$ chiral matter descending from bulk hypermultiplets, corresponding to the Lagrangian $L$ is made manifest. This approach seems to be different from the one presented in the mathematical literature [9], which roughly speaking corresponds to writing the RHS of (95) as a contribution from some boundary twisted superpotential and should be related to the use of symplectic cohomology [12–14].

[14]Note the different definition of the quantum product from [8]: $[a]_b^{\text{there}} = \epsilon^{|b|} \left[ \frac{a + \frac{\epsilon}{2}}{\epsilon} \right]_b^{\text{here}}$.

In the absence of boundary matter, the boundary module remains $\mathbb{C}[\varphi]$, and the action of bulk monopole operators on this module is a quantisation of (95)

$$\hat{v}_A = \left( \prod_\alpha \epsilon^{(-\tilde{Q}^\alpha_{A,L})_+} \left[ \frac{\hat{M}_{\alpha,L} + \frac{\epsilon}{2}}{\epsilon} \right]^{(-\tilde{Q}^A_{\alpha,L})_+} \right) \hat{p}_{-A}, \tag{100}$$

where $\hat{p}_A = e^{A_i \partial_{\varphi_i}}$. It is not hard to check that this quantisation is consistent with the quantised bulk Coulomb branch algebra (98). This yields a quantisation of the action of $\mathbb{C}[\mathcal{M}_C]$ on $QH^\bullet_T(L)$.

Using the bulk 3d theory to gauge the flavour symmetry of our 2d theory of interest $\mathcal{T}_{2d}$, the prescription to obtain the boundary module follows the same recipe as Section 4.2.2. In particular, the boundary module is enlarged to $\mathbb{C}[\sigma_a, \varphi_i]$, and the action of the Coulomb branch operators is conjugated by the integrand of the vortex partition function of $\mathcal{T}_{2d}$:

$$\hat{v}_A = \left( \prod_\alpha \epsilon^{(-\tilde{Q}^A_{\alpha,L})_+} \left[ \frac{\hat{M}_{\alpha,L} + \frac{\epsilon}{2}}{\epsilon} \right]^{(-\tilde{Q}^A_{\alpha,L})_+} \right) \left( \frac{\prod\limits_{s|q^A_s<0} \left[ \frac{M_s}{\epsilon} \right]_{-q^A_s}}{\prod\limits_{s|q^A_s>0} \left[ \frac{M_s}{\epsilon} \right]_{-q^A_s}} \right) \hat{p}_{-A}. \tag{101}$$

One again identifies:

$$\left( e^{-2\pi i \tau_a} \frac{\prod_{s|Q^a_s>0} \left[ \frac{M_s}{\epsilon} \right]_{Q^a_s}}{\prod_{s|Q^a_s<0} \left[ \frac{M_s}{\epsilon} \right]_{Q^a_s}} \hat{P}_a \right) f(\sigma, \varphi) \sim f(\sigma, \varphi), \tag{102}$$

where $\hat{P}_a : \sigma_a \mapsto \sigma_a + \epsilon$.

Note that this will not give new difference equations from the pure gauge theory case. This is because the local operators comprising the boundary module is the same, *i.e.* $\mathbb{C}[\sigma, \varphi]$ quotiented by the submodule generated by the image of (102), and the fact that for the same cocharacter $A$, the action of $\hat{v}_A$ differs from its action in the pure gauge theory case by the multiplication on the LHS by the polynomial in the first bracket of (101). Once we sandwich with the Dirichlet end of the interval, these will be set to constants (set to polynomials of the constant equivariant parameter $m$).

## Acknowledgments

It is our pleasure to thank Samuel Crew, Tudor Dimofte, Justin Hilburn, Spencer Tamagni and Ziruo Zhang for discussions.

**Funding information**  The work of AEVF was partially sponsored by the EPSRC Grant EP/W020939/1 "3d N=4 TQFTs". DZ is supported by a Junior Research Fellowship from St John's College, University of Oxford.

## A    Matrix difference equations for SQED[3]

In this Appendix, we briefly discuss how the results obtained in 4.5 apply to SQED[3], the $\mathbb{CP}^2$ sigma model, and check that we obtain results compatible with [3]. In this case the bulk theory $\mathcal{T}_{3d}$ is a pure $U(1)^2$ gauge theory, with a Coulomb branch algebra:

$$[\hat{v}_{\pm,i}, \hat{\varphi}_j] = \pm \epsilon \delta_{ij} \hat{v}_{\pm,i}, \qquad i,j = 1,2. \tag{A.1}$$

According to the derivation above, we have *e.g.*:

$$\hat{v}_{-,1} = \frac{\sigma + \varphi_1}{\sigma - \varphi_1 - \varphi_1 - \epsilon}\,\hat{p}_1\,,$$
$$\hat{v}_{-,2} = \frac{\sigma + \varphi_1}{\sigma - \varphi_1 - \varphi_1 - \epsilon}\,\hat{p}_2\,,$$

(A.2)

and identify:

$$q^{-1}(\sigma + \varphi_1)(\sigma + \varphi_1)(\sigma - \varphi_1 - \varphi_1)f(\sigma + \epsilon) \sim f(\sigma)\,,$$

(A.3)

in particular

$$\frac{1}{\sigma - \varphi_1 - \varphi_1 - \epsilon} \sim q^{-1}(\sigma + \varphi_1)(\sigma + \varphi_1)\,,$$

(A.4)

where again $q = \epsilon^3 e^{2\pi i \tau(\epsilon)}$ is the RG-invariant combination of scale and complex FI parameter.

We have that:

$$\begin{aligned}
\hat{v}_{-,1} \cdot \mathbf{1} &= \mathbf{1} + \frac{2\varphi_1 + \varphi_1 + \epsilon}{\sigma - \varphi_1 - \varphi_1 - \epsilon} \\
&\sim \mathbf{1} + (2\varphi_1 + \varphi_1 + \epsilon)q^{-1}(\sigma + \varphi_1)(\sigma + \varphi_1) \\
&= \left(1 + \varphi_1\varphi_1(2\varphi_1 + \varphi_1 + \epsilon)q^{-1}\right)\mathbf{1} \\
&\quad + (\varphi_1 + \varphi_1)(2\varphi_1 + \varphi_1 + \epsilon)q^{-1}\sigma \\
&\quad + (2\varphi_1 + \varphi_1 + \epsilon)q^{-1}\sigma^2\,.
\end{aligned}$$

(A.5)

We can do this to $\sigma$ and $\sigma^2$ to recover the rest of the matrix appearing in the difference equation (25). The results for $\hat{v}_{-,2}$ can be recovered easily by symmetry in $\varphi_1$ and $\varphi_1$. Overall, we obtain full agreement with the results derived from different methods in [3].

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
