# Peer review of "Difference Equations: from Berry Connections to the Coulomb Branch"

_SciPost Physics, doi:SciPost Phys. 18, 045 (2025)_

## Round 1 · Referee Report · Anonymous (Referee 1) · 2024-11-29

Strengths

In previous work, the authors had pointed out a connection between certain 2D GLSMs, spectral data of monopole solutions and difference equations satisfied by B-brane amplitudes. In the present article, the authors further analyse this by establishing a connection to 3D theories where the relevant structures arise on the Coulomb branch of an N=4 abelian gauge theory. The authors give a construction that relates the 2D and 3D setups, show that the relevant data emerges within the 3D theory and give non-trivial evidence via explicit examples.

The article succeeds in clarifying connections between different mathematical structures via the physics of supersymmetric gauge theories. It adds new conceptual understanding and touches on various topical research areas such as the interplay between topological structures in supersymmetric quantum field theories in various dimensions, difference equations and the associated mathematics, quantum cohomology and quantum K-theory, and D-branes in supersymmetric field theories.

Weaknesses

The article is generally of a high quality. One potential weakness is that the authors focus only on abelian theories in their examples. At various places in the article the authors claim that generalisations to non-abelian models are straight-forward. If this indeed the case, it would have been nice to go beyond SQED and discuss a simple non-abelian example. However, non-abelian GLSMs tend to have a very rich structure that may lend itself to a more detailed analysis in a separate article. I would like to encourage the authors to look into this.

Report

This is a well-written article that provides a new conceptual understanding of structures that are of interest in mathematical physics and pure mathematics. To my knowledge, the results are new and provide interesting directions for further research. While the authors could have gone a bit further in their analysis of concrete examples, the article certainly contains enough new results to warrant a publication in SciPost. Publication in SciPost is therefore recommended.

Requested changes

The article does not require any major revisions, but the authors may consider correcting the following typos before sumbmitting the final version of their article.

1) p.5, line 2: characterise -> characterised
2) p.6, below (2.19): Bogomonly -> Bogomolny
3) p.9, below (3.2): values -> valued
4) p.9, eq. (3.4) and two lines above: When referring to the left boundary, do the authors actually mean $t=0$?
5) p.14, first line: close the parenthesis at $l(\mathcal{O}^{\mathrm{2d}}$
6) p.14, two lines above (4.11): correct $bra\mathcal{D}_m$

Recommendation

Publish (easily meets expectations and criteria for this Journal; among top 50%)

---

## Round 1 · Referee Report · Anonymous (Referee 2) · 2024-12-22

Report

This is continuation of previous work of the authors about spectral varieties for the Berry connection of 2D GLSMs and their relation to N=4 3D Coulomb branches. The authors explore the connection by applying a sandwich construction relating the 2D and 3D setups by placing the 2D theory on the boundary and gauging the flavor symmetry. The construction is demonstrated with examples, mainly SQED with N flavors.
This is an highly techical article concerning the interplay between various mathematical structures recently discussed in the context of supersymmetric gauge theories. The results are sound and potentially useful and the paper meets the SciPost standards of quality.

Recommendation

Publish (meets expectations and criteria for this Journal)

---

## Editorial Decision

published